# Prognostic Value of Resistance Proteins in Plasma Cells from Multiple Myeloma Patients Treated with Bortezomib-Based Regimens

**DOI:** 10.3390/jcm10215028

**Published:** 2021-10-28

**Authors:** Paweł Robak, Janusz Szemraj, Damian Mikulski, Izabela Drozdz, Karolina Juszczak, Dariusz Jarych, Małgorzata Misiewicz, Kacper Kościelny, Wojciech Fendler, Tadeusz Robak

**Affiliations:** 1Department of Experimental Hematology, Medical University of Lodz, 93-510 Lodz, Poland; pawel.robak@umed.lodz.pl; 2Department of Medical Biochemistry, Medical University of Lodz, 92-215 Lodz, Poland; janusz.szemraj@umed.lodz.pl (J.S.); karolina.juszczak@stud.umed.lodz.pl (K.J.); 3Department of Biostatistics and Translational Medicine, Medical University of Lodz, 92-215 Lodz, Poland; damian.mikulski@stud.umed.lodz.pl (D.M.); kacper.koscielny@stud.umed.lodz.pl (K.K.); wojciech.fendler@umed.lodz.pl (W.F.); 4Department of Clinical Genetics, Medical University of Lodz, 92-213 Lodz, Poland; izabela.drozdz@umed.lodz.pl; 5Laboratory of Personalized Medicine, Bionanopark, 93-465 Lodz, Poland; djarych@cbm.pan.pl; 6Laboratory of Virology, Institute of Medical Biology, Polish Academy of Sciences, 93-232 Lodz, Poland; 7Department of Hematology, Medical University of Lodz, 93-510 Lodz, Poland; malgorzata.misiewicz@umed.lodz.pl; 8Copernicus Memorial Hospital, 93-513 Lodz, Poland

**Keywords:** bortezomib, cMAF, MAFb, multiple myeloma, POMP, prognosis, PFS, survival, OS

## Abstract

While multiple myeloma (MM) treatment with proteasome inhibitors and other agents yields encouraging results, primary and secondary resistance remains an emerging problem. An important factor in such treatment resistance is the overexpression of several proteins. The present study comprehensively evaluates the expression of POMP, PSMB5, NRF2, XBP1, cMAF and MAFb proteins in plasma cells isolated from the bone marrow of 39 MM patients treated with bortezomib-based regimens using an enzyme-linked immunosorbent assay (ELISA). The proteins were selected on the basis of previous laboratory and clinical studies in bortezomib-treated MM patients. It was found that the expression of the investigated proteins did not significantly differ between bortezomib-sensitive and bortezomib-refractory patients. However, the expression of some proteins correlated with overall survival (OS); this was significantly shorter in patients with higher POMP expression (HR 2.8, 95% CI: 1.1–7.0, *p* = 0.0277) and longer in those with higher MAFB expression (HR 0.32, 95% CI: 0.13–0.80, *p* = 0.0147). Our results indicate that a high expression of POMP and MAFB in MM plasma cells may serve as predictors of OS in MM patients treated with bortezomib-based regimens. However, further studies are needed to determine the role of these factors in effective strategies for improving anti-myeloma therapy.

## 1. Introduction

Multiple myeloma (MM, plasma cell myeloma) is a hematological malignancy characterized by the accumulation of malignant plasma cells (PC) in the bone marrow (BM), often resulting in bone lesions, hypercalcemia, infections, anemia and the production of monoclonal immunoglobulin [1]. The disease accounts for about 1.8% of all cancers and 18% of all hematologic malignancies, with an annual incidence of 4.5–6 cases per 100,000 [2,3]. The median age at diagnosis is 69 years and median survival is around four or five years (Stat Fact Sheets SEER. Myeloma. http://seer.cancer.gov/statfacts/html/mulmy.html, accessed on 10 June 2021).

The proteasome inhibitors (PI), bortezomib, carfilzomib and ixazomib play a key role in the treatment of MM, and several new drugs from this group are undergoing clinical trials [4,5]. These agents are reversible inhibitors of the 26S proteasome, which plays a critical role in the pathogenesis and proliferation of MM cells. Proteasome inhibitors act via various mechanisms, including by exerting direct effects on myeloma cells, and inhibiting the activity of cytokines as well as several adhesion molecules and angiogenesis. They are also known to inhibit the action of nuclear factor kappa B (NF-κB), which plays a key role in the survival and proliferation of MM cells. Proteasome inhibition causes a number of downstream effects, including the inhibition of NF-κB signaling [6]. The resulting endoplasmic reticulum stress leads to an unfolded protein response, the downregulation of growth factor receptors, suppression of adhesive molecule expression and inhibition of angiogenesis.

Bortezomib is a boronic acid dipeptide that reversibly inhibits the β5 subunit of proteasome and thus its chymotrypsin-like activity [7,8]. Bortezomib is approved for first-line treatment and in relapsed/refractory patients [9]. However, eventually after several courses of treatment, most patients show resistance to bortezomib, and most demonstrate multiple drug resistance. In addition, approximately 20% of patients exhibit primary resistance that determines a lack of response to treatment [10,11]. Reports indicate overexpression of several proteins in MM plasma cells, and that this may be an important factor in the pathomechanism of bortezomib resistance [11,12,13,14]. In addition, proteomics-based techniques have highlighted the proteomic patterns specific to response and treatment outcome in MM patients, and have created a possibility for the implementation of marker-based individualized therapies [15,16].

The present study performs a comprehensive evaluation of the expression of six previously described proteins in PCs, which may influence the prognosis in MM patients. Samples were taken from the BM of 39 MM patients treated with bortezomib-based regimens and the expression of the POMP, PSMB5, NRF2, XBP1, cMAF and MAFB proteins was determined using enzyme-linked immunosorbent assay (ELISA). The proteins were selected on the basis of previous laboratory and clinical studies investigating their expression in bortezomib-sensitive and refractory MM patients and their influence on the treatment results of bortezomib-based regimens. Their characteristics are presented in Table 1.

## 2. Materials and Methods

### 2.1. Patients

A total of 39 MM patients treated at the Department of Hematology, Copernicus Memorial Hospital, Lodz, Poland, were included in the study. Their demographic, clinical and laboratory details are shown in Table 1. The patients were classified as either bortezomib-sensitive or bortezomib-refractory, according to their response to bortezomib-based therapy [32,33]. Response to treatment and relapse/progression were classified based on International Myeloma Working Group (IMWG) criteria [34,35]. The patients’ refractory to bortezomib-based treatment demonstrated progressive MM during bortezomib therapy or at two months from the end of treatment [34,35,36]. The study was conducted according to good clinical and laboratory practice rules and the principles of the Declaration of Helsinki. The study protocol was approved by the ethics committee of the Medical University Lodz (No RNN/103/16/KE). Written informed consent was obtained from all patients for anonymous usage of their diagnostic routine sample for the scientific project.

### 2.2. Collection of MM Cells

Bone marrow samples were collected from 39 MM patients before treatment with a bortezomib-based regimen. Plasma cells were isolated from fresh BM aspirate samples using CD138 microbeads (Miltenyi Biotec Inc., Bergisch Gladbach, Germany). Purity was verified by flow cytometry on the basis of CD38 and CD138 expression, and was higher than 80% for all samples. Purified plasma cells from all studied patients were stored in DMEM medium with 10%DMSO 10% FBS at −80 °C for further use.

### 2.3. Determination of Human Protein Level in Plasma Cells

The expression of six human proteins (POMP, PSMB5, NRF2, XBP1, cMAF and MAFB in MM PCs was analyzed in all samples. Quantitative detection of human protein concentration in PCs was performed by solid-phase sandwich ELISA assay, as previously described [37,38]. POMP (Biobool, Hong Kong), PSMB5 (MyBioSource, San Diego, CA, USA), NRF2 (ThermoFischer Scientific, Waltham, MA, USA), XBP1 (MyBioSource, USA), cMAF (ThermoFischer Scientific, USA) and MAFb (MyBioSource, USA) reagents were applied. Lysates from PC pellets were prepared using Qiagen All Prep DNA/RNA/protein kit, according to the manufacturer’s protocol. Up to 7 × 10^6^ cells were used for the isolation. After the DNA and RNA isolation, protein pellet was suspended in PBS (1×) buffer. The total protein concentration was determined by BCA kit; this value should not exceed 0.3 mg for each sample. All samples were added to the appropriate wells, as well as standards to create a standard curve, with immobilized antibodies specific for human POMP, PSMB5, NRF2, XBP1, cMAF and MAFB proteins, and incubated. All standards and samples were pipetted in triplicate to the plate. After incubation and washing, biotin-conjugated anti-human POMP, PSMB5, NRF2, XBP1, cMAF and MAFB antibodies were added and incubated. After washing away any unbound substances, biotinylated antibody, the horseradish peroxidase-conjugated streptavidin was pipetted to the wells. Unbound immunoglobulins were washed off. An enzyme-labeled anti-human globulin was then bound to the antigen antibody complex. After washing, the bound conjugate was developed with the aid of a substrate solution (TMB) to render a blue soluble product, which turns yellow after adding the acid-stopping solution. The optical density (OD) was measured at 450 nm with wavelength correction at 620 nm, by using a Thermo LabSystems Multiskan Ascent 354 Plate Reader (LabX, Midland, Canada).

A standard curve was generated using curve-fitting software. The standard curve was plotted as the relative OD450 of each standard solution (Y) vs. the respective concentration of the standard solution (X). The target concentration of the samples, i.e., the relative OD 450, calculated as (the OD 450 of each well)—(the O D 450 of blank well), and was interpolated from the standard curve. The concentrations of the samples were calculated by the appropriate dilution factor.

### 2.4. Statistical Analysis

Continuous variables were presented as mean ± standard deviation (SD). For normally distributed data, the relationships between two groups of continuous variables were determined using the two-sided independent Student’s t-test. Survival analysis was conducted using a Kaplan–Meier estimate with univariate and multivariate Cox’s proportional hazards models, as well as the log-rank test. Examined protein level was divided by ACTB protein level to normalize values. Cutoff Finder was used to determine the optimal cut point for protein level [39]. The optimal cutoff determined by Cutoff Finder is defined as the point with the most significant (log-rank test) split for OS. For clinics, where most of the decisions are binary, it seems most appropriate to use a procedure based on the stratification of a continuous biomarker variable into two groups. Normalized protein expression values for all samples and the most important clinical variables were provided in Appendix A.

## 3. Results

### 3.1. Characteristics of the Patients Included in the Analysis

The demographic, clinical and laboratory characteristics of the MM patients enrolled for the study are presented in Table 2. The mean age of the study cohort was 66.8 ± 8.9 years (range: 39–81). Ten patients had received at least one prior therapy before bortezomib-based regimen initiation. Twenty-three (59.0%) patients displayed IgG paraprotein, eight (20.5%) demonstrated IgA and nine (23.1%) had light chain disease (LCD). Cytogenetics data were available for 20 patients (51.2%): amp (1q) was the most common abnormality (28.2%), followed by IGH rearrangements (17.9%), followed by t(4;14) (10.3%), del(13q) (5.1%) and del(17p) (5.1%).

Most of the patients (76.9%) had received a VCD (bortezomib, cyclophosphamide and dexamethasone) regimen, four (10.3%) VMP (bortezomib, melphalan and prednisone), one (2.6%) VTD (bortezomib, thalidomide and dexamethasone) and four VD (bortezomib and dexamethasone). Eighteen patients (46.2%) underwent ASCT. Data on the response according to IMWG criteria to bortezomib-based therapy were available for 39 (100%) patients. Ten patients achieved CR (25.6%), ten (25.6%) a very good partial response (VGPR) and eight (20.5%) a partial response (PR). Overall, 27 of the 39 patients were bortezomib-sensitive and 12 were refractory to bortezomib-based regimens.

### 3.2. Protein Levels According to Clinical and Laboratory Characteristics

Comparisons of protein level according to International Staging System (ISS), previous treatment and CRAB symptoms (anaemia, hypercalcaemia, renal failure and osteolytic bone lesions) are provided in Appendix A. Generally, higher levels of cMAF (*p* = 0.0383) were observed in patients with ISS stage III compared to stages I and II (Figure 1A, Appendix A). POMP level was higher in patients without previous treatment (*p* = 0.0350, Figure 1B, Appendix A). No significant difference was found between selected protein levels with regard to CRAB symptoms: anemia at diagnosis, hypercalcemia, renal failure, bone disease and bortezomib refractoriness (Appendix A). NRF2 level was increased in patients with CR or VGPR response after bortezomib-based therapy compared to those with a worse response (*p* = 0.0167, Figure 1C, Appendix A). No statistically significant differences in protein expression levels were observed between the bortezomib-refractory and bortezomib-sensitive groups (Appendix A).

### 3.3. Influence of ASCT and ISS Protein Levels on Overall Survival and Progression-Free Survival

Data on progression-free survival (PFS) and overall survival (OS) was available for all patients. The median PFS was 11.6 (95% CI: 8.1–13.3) months and the median OS was 28.2 (95% CI: 21.0–34.2) months. Univariate Cox proportional hazards regression analysis was conducted to determine the prognostic value of the clinical variables and the normalized protein levels. The optimal cut points for normalized protein level dichotomization, determined by Cutoff Finder, are provided in Appendix A.

ISS III (HR 2.12, 95% CI: 1.07–4.19, *p* = 0.0308) and previous treatment (HR 2.65, 95% CI 1.18–5.91, *p* = 0.0178) were associated with shorter PFS (Table 3). Autologous stem cell transplant (ASCT) in the treatment schedule (HR 0.35, 95% CI: 0.17–0.71, *p* = 0.0035) and at least very good partial response (≥VGPR) to myeloma treatment (HR 0.47, 95% CI: 0.24–0.93, *p* = 0.0292) were related with longer PFS. Protein levels did not impact PFS significantly. Statistically significant variables were entered to multivariate analysis using Cox’s proportional hazards regression model with a stepwise selection procedure. The final model consisted of two variables: ≥VGPR and ASCT (Table 3).

The only clinical variable that significantly impacted OS was ASCT procedure (HR 0.27, 95% CI: 0.10–0.70, *p* = 0.0074). Among the proteins that significantly impacted OS in univariate analyses, high expression of POMP (HR 2.8, 95% CI: 1.1–7.0, *p* = 0.0277) was related with shorter OS, whereas high expression of MAFB (HR 0.32, 95% CI: 0.13–0.80, *p* = 0.0147) was associated with longer OS. The corresponding Kaplan–Meier plots are shown in Figure 2. Significant variables were entered to multivariate analysis using Cox’s proportional hazards regression model with a stepwise selection procedure. The final multivariate model included all three significant variables: ASCT, high POMP expression and high MAFB expression (Table 4).

## 4. Discussion

The present study examined the levels of six proteins (POMP, PSMB5, NRF2, XBP1, cMAF and MAFB) in the PC from BM of 39 MM patients, treated with bortezomib-based regimens. ELISA was used to detect cellular protein levels. None of the investigated proteins significantly differed between bortezomib-sensitive and bortezomib-refractory patients. However, the expression of some proteins correlated with OS. In particular, OS was significantly shorter in patients with higher expression of POMP (*p* = 0.0277) and longer in patients with higher expression of MAFB (*p* = 0.0147). The median OS for all patients in our study was only 28.2 months. This value is significantly shorter than that given in the recant American SEER data, where the five-year relative survival rate is 54% (Stat Fact Sheets SEER. Myeloma. http://seer.cancer.gov/statfacts/html/mulmy.html, accessed on 10 June 2021). This may be due to the inclusion of more refractory patients in our group and differences in the treatment. For example, the SEER data included patients with localized disease (solitary plasmacytoma) and who were more likely to be receiving novel drugs (karfilzomib, pomalidomid and daratumumab). In addition, our cohort mainly consisted of patients requiring an inpatient approach and who were diagnosed during hospitalization. Our group demonstrated a high incidence of ISS III (44%) and bone disease (51%), which are known to significantly impair treatment outcome. Furthermore, 25% of our patients had previously been treated with at least one prior therapy before the bortezomib-based regimen initiation.

The MAFB and cMAF proteins are two key members in the MAF family. They share a similar structure, acting as transcription factors regulating gene transcription by cyclic adenosine monophosphate–response elements, specifically those including a DNA-binding domain and transcription activation domain [26]. cMAF is a critical oncogenic transcription factor influencing myelomagenesis [27]. Recent studies suggest that cMAF and MAFB proteins are substrates of USP5 (Ubiquitin specific peptidase 5) [28]. High MAFB protein expression is one of the most frequent oncogenic events in the progression of MM, and confers innate resistance to bortezomib [30]. Quiang et al. demonstrated that high MAFB protein expression is associated with resistance to proteasome inhibitors, bortezomib and carfilzomib in MM cell lines, and that MAFB mediated proteasome inhibitor resistance [30]. However, in the present study, similar levels of MAFB were observed in BM plasma cells of bortezomib-sensitive and bortezomib-refractory patients.

Low expression of POMP in MM cells was also associated with longer OS. The POMP expression is essential for de novo biogenesis of proteasome, and its increased expression is one of the mechanisms of acquired resistance to PI [17]. There are also reports demonstrating that some cell lines resistant to bortezomib have increased expression of POMP [12,17]. It has been previously documented that serum POMP mRNA was significantly upregulated in MM patients’ refractory to bortezomib-based treatment [40]. However, in the present study, bortezomib-sensitive and refractory patients demonstrated similar levels of POMP expression in MM at the protein level, probably due to the relatively low number of patients.

In a study of cell lines (V10R) RPMI 8226, OPM-2, ANBL-6 and KAS-6/1, Li et al. report an increase in POMP protein expression in MM cells resistant to bortezomib [17]; they also found the suppression of POMP protein via shRNAs to restore cell sensitivity, while its over-expression in cells not subject to prior treatment increased resistance. The researchers also identified a protein-binding site for a suppressive factor, Nuclear factor erythroid 2-related factor 2 (NRF2) in the promoter region of the POMP protein. Although its increased expression should increase sensitivity to bortezomib, increased NRF2 levels were observed in resistant cells, together with increased levels of POMP protein. The level of activation of the two proteins varied between cell lines, and it appeared to have a stronger effect on bortezomib sensitivity in the KAS-6/1 than the OPM-2 line [17].

It was not possible to document the prognostic value of other proteins, including PSMB5, NRF2 and XBP1. The proteasome subunit β5 (PSMB5) is a protein encoded by the PSMB5 gene that contributes to the complete assembly of 20S proteasome complex. PSMB5 point mutations and β5 subunit overexpression were the most frequent changes observed in the bortezomib-resistant cell lines [18,19,20]. Nuclear factor erythroid 2-related factor 2 (NRF2) is a key regulator of MM cell survival in patients treated with PI [22]. NRF2 is constitutively activated in 50% of samples from MM patients, as well as several MM cell lines [22]. In addition, genetic inhibition of constitutively-expressed NRF2 reduced the viability of MM cells. Importantly, PI increased the expression of NRF2 both in primary MM cells and in MM cell lines. Finally, inhibition of NRF2 in combination with PI treatment significantly increases apoptosis in MM cells [22]. XBP1 is an important transcription factor necessary for B cell differentiation into PCs, being responsible for the final maturation of plasmablasts to plasmocytes and the induction of immunoglobulin secretion [41]. In a recent study, Brojan et al. found that bortezomib-resistant cells and BTZ-refractory MM patients exhibited lower sXBP1 levels [24]. These observations suggest that determination of sXBP1 levels prior to bortezomib treatment in MM may be useful to predict bortezomib resistance; however, these results were not confirmed in our present study.

The prognostic value of proteomic profiling has previously been reported by other authors [14,15]. Rajpal et al. used ELISA to validate the candidate protein biomarkers using unfractionated serum from 51 newly diagnosed MM patients subsequently treated with thalidomide-based regimens [16]. Recently, Bai et al. analyzed the feasibility of predicting response to thalidomide-based therapy in previously untreated MM using a novel panel of predictive serum markers; they used serum proteomic profiling to construct a MM model using four selected peptides evaluated in different disease states [42]. Western blot and ELISA were employed to validate the variability. The results demonstrate that a proteomics-based approach using a combination of immunodepletion, 2D-difference gel electrophoresis analysis, mass spectrometry and ELISA is an effective strategy for identifying proteins useful for predicting response to thalidomide. It was found that ZAG, VDB, SAA, B2M and Hp were demonstrated significantly different serum concentrations between thalidomide-refractory and thalidomide-sensitive patients.

However, in contrast to our present study, those given above used serum for proteomic profiling. In contrast, Dytfeld et al. performed a proteomic analysis of the PCs of MM patients subsequently treated with a proteasome inhibitor [15]. Their findings identify proteomic signatures that can be used to differentiate patients who achieved at least VGPR from those with a lower response to bortezomib-based chemotherapy. However, the study did not address any of the proteins included in the present analysis.

## 5. Conclusions

Lower expression of POMP and higher expression of MAFB correlated with longer OS in MM patients treated with borteomib. Evaluation of these proteins in plasma cells can be potentially useful in MM patients treated with bortezomib-based regimens. However, the clinical and biological significance of these findings needs further investigation.

## Figures and Tables

**Figure 1 jcm-10-05028-f001:**
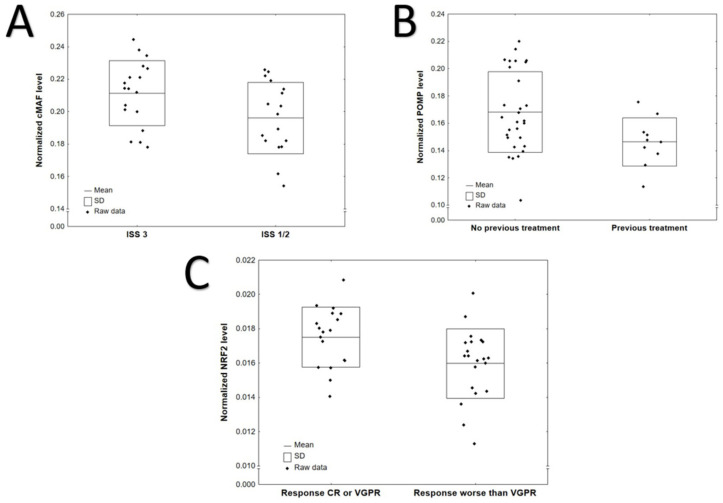
Significant differences in protein levels according to: (**A**) ISS: cMAF, *p* = 0.0383; (**B**) previous treatment: POMP, *p* = 0.0350 (**C**) response after bortezomib-based treatment: NRF2, *p* = 0.0167.

**Figure 2 jcm-10-05028-f002:**
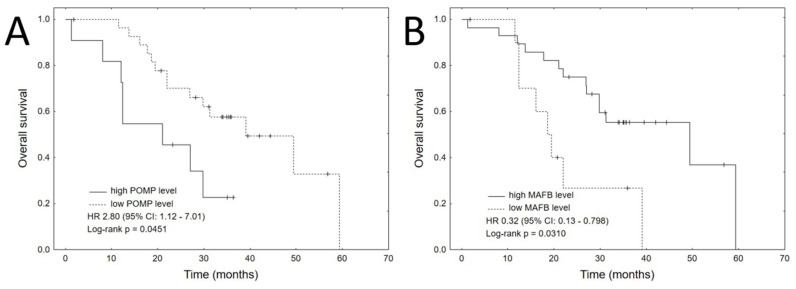
Kaplan–Meier plots for each of the significant proteins in the univariate analyses for OS: (**A**) POMP, (**B**) MAFB.

**Table 1 jcm-10-05028-t001:** The characteristics of investigated proteins and their significance in MM.

Protein	Characteristics	Significance in MM	Reference
POMP	A short-lived maturation factor essential for 20S proteasome subunit biogenesis.	POMP over-expression contributes to proteasome inhibitor resistance, while suppression enhances bortezomib and carfilzomib activity.	[12,17]
PSMB5	A component of the 20S core proteasome complex involved in the proteolytic degradation of most intracellular proteins.	Overexpression observed in bortezomib-resistant cell lines, PSMB5 contributes to bortezomib resistance in MM patients.	[18,19,20,21]
NRF2	A transcription activator that binds to antioxidant response elements in the promoter regions of target genes.	A key regulator of MM survival in treatment naive and PI-treated cells, PI increases expression of NRF2 in MM cells.	[22,23]
XBP1	A transcription factor found during endoplasmic reticulum stress; a regulator of the UPR; needed for differentiation of B cells into PCs.	XBP1 levels correlate with bortezomib resistance in MM; XBP1 levels are low in bortezomib-refractory MM patients	[24,25]
cMAF	A bZIP zipper transcription factors, belonging to the AP-1 family.	Overexpressed in MM, enhancing tumor-stroma interactions.	[26,27,28,29]
MAFB	bZIP transcription factor that plays an important role in the regulation of lineage-specific hematopoiesis.	High expression is associated with resistance to proteasome inhibitors, frequent event in the progression of MM.	[26,27,28,30,31]

Abbreviations: bZIP—basic leucine zipper; USP5—Ubiquitin specific peptidase 5; MM—multiple myeloma; POMP—proteasome maturation protein; PSMB5—proteasome subunit β5; NRF2- nuclear factor erythroid 2-related factor 2; PC—plasmacells, PI—proteasome inhibitor; UPR—unfolded protein response, XBP1—X-box-binding protein 1.

**Table 2 jcm-10-05028-t002:** The characteristics of the MM patients included in the analysis. Data shown as frequencies N (%), unless otherwise specified.

Characteristics	Overall (N = 39)
Sex	M: 23 (59.0)
	F: 16 (41.0)
Age mean ± SD	66.8 ± 8.9
(range)	(39–81)
ISS at diagnosis	I: 12 (30.8)
	II: 7 (17.9)
	III: 17 (43.6)
	Data missing: 3 (7.7)
Paraprotein	
IgG	23 (59.0)
LCD	8 (20.5)
IgA	8 (20.5)
HB < 10 g/dL at diagnosis	14 (35.9)
Creatinine > 2 mg/dL at diagnosis	4 (10.3)
Calcium > 2.5 mmol/L at diagnosis	11 (28.2)
Bone disease	20 (51.3)
CRP > 5 mg/L	16 (41)
LDH > 240 U/L	10 (25.6)
Cytogenetics *	
amp(1q)	11 (28.2)
t(4;14)	4 (10.3)
del(13q)	2 (5.1)
del(17p)	2 (5.1)
t(11;14)	1 (2.6)
del(1p)	1 (2.6)
t(14;16)	0 (0)
t(14;20)	0 (0)
IGH rearrangements	7 (17.9)
Prior treatment	10 (25.6)
Bortezomib regimen:	
VCD	30 (76.9)
VMP	4 (10.3)
VD	4 (10.3)
VTD	1 (2.6)
ASCT	18 (46.2)
RTx	10 (25.6)
Response to treatment	
CR	10 (25.6)
VGPR	10 (25.6)
PR	8 (20.5)
SD	5 (12.8)
PD	6 (15.4)
Refractoriness to bortezomib	12 (30.8)

* Cytogenetics data were available for 20 patients (51.2%). Abbreviations: LCD—light chain disease; VCD—bortezomib, cyclophosphamide and dexamethasone; VMP—bortezomib, melphalan and prednisone; VTD—bortezomib, thalidomide and dexamethasone; VD—bortezomib and dexamethasone; ASCT—autologous steam cell transplantation, RTx—radiotherapy and ISS- International Staging System.

**Table 3 jcm-10-05028-t003:** Univariate Cox regression analyses for progression-free survival and overall survival.

Variables	PFS					OS				
	Coefficient	*p*-Value	HR	95% CI		Coefficient	*p*-Value	HR	95% CI	
				Lower	Higher				Lower	Higher
ISS III	0.38	0.0308	2.12	1.07	4.19	0.34	0.1284	1.95	0.82	4.64
Previous treatment	0.49	0.0178	2.65	1.18	5.91	0.16	0.5265	1.36	0.52	3.56
≥VGPR	−0.38	0.0292	0.47	0.24	0.93	−0.18	0.4102	0.69	0.29	1.66
ASCT	−0.53	0.0035	0.35	0.17	0.71	−0.65	0.0074	0.27	0.10	0.70
Sex (M)	0.12	0.4781	1.28	0.65	2.53	0.05	0.8298	1.10	0.45	2.69
HB < 10 g/dL	0.12	0.4967	1.27	0.64	2.54	0.07	0.7431	1.16	0.48	2.81
Calcium > 2.5 mmol/L	−0.13	0.4838	0.77	0.38	1.59	0.01	0.9747	1.02	0.40	2.55
Creatinine > 2 mg/dL	−0.14	0.6125	0.76	0.27	2.17	0.22	0.482	1.57	0.45	5.46
Bone disease	0.03	0.8753	1.06	0.54	2.06	−0.06	0.7691	0.88	0.37	2.08
High POMP	−0.064	0.7434	0.880	0.410	1.889	0.515	0.0277	2.802	1.120	7.010
High PSMB5	0.059	0.7343	1.125	0.570	2.219	0.271	0.2319	1.720	0.707	4.188
High NRF2	−0.205	0.2784	0.663	0.315	1.394	−0.264	0.3104	0.590	0.213	1.635
High XBP1	−0.047	0.7954	0.911	0.450	1.845	−0.301	0.2419	0.548	0.200	1.501
High cMAF	0.237	0.1972	1.608	0.781	3.310	0.280	0.0988	2.522	0.841	7.565
High MAFB	−0.270	0.1582	0.583	0.276	1.233	−0.572	0.0147	0.319	0.127	0.798

**Table 4 jcm-10-05028-t004:** Final multivariate Cox regression analyses for PFS and OS of MM patients.

Variables	PFS				
	Coefficient	*p*-Value	HR	95% CI	
				Lower	Higher
≥VGPR	−0.43	0.0170	0.43	0.21	0.86
ASCT	−0.56	0.0021	0.32	0.16	0.66
	**OS**				
ASCT	−0.79	0.0025	0.20	0.07	0.57
High MAFB	−0.92	0.0005	0.16	0.06	0.45
High POMP	0.60	0.0189	3.30	1.22	8.94

## Data Availability

Normalized protein expression values for all samples and the most important clinical variables were provided in Appendix A. More data presented in this study are available from the corresponding author on request.

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
