# Peer review of "Prognostic Value of Resistance Proteins in Plasma Cells from Multiple Myeloma Patients Treated with Bortezomib-Based Regimens"

_jcm, 2021, doi:10.3390/jcm10215028_

Round 1

Reviewer 1 Report

This paper is somewhat confusing especially since the authors do not disclose why they chose the 6 proteins that they chose to evaluate until the discussion section. For clarity, this should probably be moved to the introduction section. Additionally, the authors are looking for bortezomib inherit (primary) refractory disease when really this represents a very small proportion of patients. The vast majority of patients acquire resistance to bortezomib and it may be more clinically relevant to investigate changes in the proteins in patients pre and post bortezomib refractoriness.  Additionally, the various bortezomib containing combinations tested in this manuscript further complicates the data as by definition patient that did not respond to these therapies would be primary refractory to all these agents which predicts a significantly worse phenotype and is likely why response correlates with ISS stage and ASCT. Finally, it is particularly troubling that the patients included in this study only had an OS of 28.2 months. The authors quote US data (using the SEER database and Cancer statistics) and do not clarify that this data only represents the US experience, but they should know that the OS in the US is significantly better then 28.2 months. This either represents a more refractory patient sample or some issue with their treatment that caused them to have a particularly poor prognosis. This should at least be discussed in the manuscript. 

Author Response

Reviewer #1:  This paper is somewhat confusing especially since the authors do not disclose why they chose the 6 proteins that they chose to evaluate until the discussion section.

For clarity, this should probably be moved to the introduction section.

Response: We moved this paragraph   the introduction section: “The proteins were selected on the basis of previous laboratory and clinical studies investigating their expression in bortezomib-sensitive and refractory MM patients and their influence on the treatment results of bortezomib-based regimens”.

Additionally, the authors are looking for bortezomib inherit (primary) refractory disease when really this represents a very small proportion of patients. The vast majority of patients acquire resistance to bortezomib and it may be more clinically relevant to investigate changes in the proteins in patients pre and post bortezomib refractoriness. 

Response: We agree with the Reviewer regarding the investigation of  changes in the proteins in patients pre and post bortezomib refractoriness. Unfortuntely, this data is not currently available; however, we are planning to investigate these changes in the future.  Although a functional analysis of how these proteins change their activity within cells during different phases of treatment would be an exciting study to perform, it would likely require a different experimental model, cell cultures, and in-depth mechanistic evaluations far exceeding the scope of this survival-oriented analysis.

 Additionally, the various bortezomib containing combinations tested in this manuscript further complicates the data as by definition patient that did not respond to these therapies would be primary refractory to all these agents which predicts a significantly worse phenotype and is likely why response correlates with ISS stage and ASCT.

Response: We agree with the reviewer. However, bortezomib monotherapy is rarely used. We have modified the tile for “Prognostic value of resistance proteins in plasma cells from multiple myeloma patients treated with bortezomib based regimens "

 Finally, it is particularly troubling that the patients included in this study only had an OS of 28.2 months. The authors quote US data (using the SEER database and Cancer statistics) and do not clarify that this data only represents the US experience, but they should know that the OS in the US is significantly better then 28.2 months. This either represents a more refractory patient sample or some issue with their treatment that caused them to have a particularly poor prognosis. This should at least be discussed in the manuscript. 

Response: We have added the proper comment in the Discussion section : “The median OS for all patients in our study was only 28.2 months. This value is significantly shorter than that given in the recant American SEER data, where the five-year relative survival rate is 54% (Stat Fact Sheets SEER. Myeloma. http://seer.cancer.gov/statfacts/html/ mulmy.html. Accessed on June 10th, 2021). This may be due to inclusion of more refractory patients in our group and differences in the treatment. For example, the SEER data included patients with localized disease (solitary plasmacytoma) and who were more likely to be receiving novel drugs (karfilzomib, pomalidomid, daratumumab). In addition our cohort was mainly consisted of patients requiring an inpatient approach and who were diagnosed during hospitalization. Our group demonstrated a high incidence of ISS III (44%) and bone disease (51%), which are known to significantly impair treatment outcome. Furthermore, 25% of our patients had previously been treated with at least one prior therapy before bortezomib-based regimen initiation”    

Reviewer 2 Report

The authors have performed a study aiming to evaluate the prognostic value of resistance proteins in plasma cells from multiple myeloma patients treated with bortezomib.

The paper is written in a good English, and it can be clear for readers.

The paper is complete, well organized, and can be really interesting for readers.

A table with data from literature about the main proteins could be useful for readers.

The idea is good and well presented and the paper could be really interesting for readers, and it can be accepted after minor revision.

Author Response

Reviewer #2:

 The authors have performed a study aiming to evaluate the prognostic value of resistance proteins in plasma cells from multiple myeloma patients treated with bortezomib.

The paper is written in a good English, and it can be clear for readers.

The paper is complete, well organized, and can be really interesting for readers.

A table with data from literature about the main proteins could be useful for readers.

The idea is good and well presented and the paper could be really interesting for readers, and it can be accepted after minor revision.

Response: We thank the reviewer for the positive review of our paper. We have added a further table (Table 1) with data from the literature about the main proteins involved in MM.

Reviewer 3 Report

Authors evaluate the expression of POMP, PSMB5, NRF2, XBP1, cMAF and MAFb in plasma cells using ELISA method and concluded that high expression of POMP and MAFb in MM plasma cells may be related to OS in MM patients treated with bortezomib-based regimens.

Unfortunately, this manuscript is a different version of authors recently published paper in Cancers in February 2021(“The Prognostic Value of Whole-Blood PSMB5, CXCR4, POMP, and RPL5 mRNA Expression in Patients with Multiple Myeloma Treated with Bortezomib”), with the difference in mRNA level vs. protein expression levels.

Authors state in the abstract about the expression of POMP, PSMB5, NRF2, XBP1, cMAF and MAF, but in results they only show POMP, NRF2, and MAFb. Abstract needs to be fixed since authors focus are these 3 proteins.  

Results section needs to be organized into subsections and have its own results, so that readers would follow the manuscript better.

Author Response

Reviewer #3:

Authors evaluate the expression of POMP, PSMB5, NRF2, XBP1, cMAF and MAFb in plasma cells using ELISA method and concluded that high expression of POMP and MAFb in MM plasma cells may be related to OS in MM patients treated with bortezomib-based regimens.

Unfortunately, this manuscript is a different version of authors recently published paper in Cancers in February 2021("The Prognostic Value of Whole-Blood PSMB5, CXCR4, POMP, and RPL5 mRNA Expression in Patients with Multiple Myeloma Treated with Bortezomib"), with the difference in mRNA level vs. protein expression levels.

Response: We would like to thank the Reviewer for these suggestions. In fact this report is the continuation of a previous study published recently in Cancers; this study assessed the expression of the previously-described genes at the mRNA level in the peripheral blood. In the present study we evaluated the expression of POMP, PSMB5, NRF2, XBP1, cMAF and MAFb proteins in plasma cells isolated from the bone marrow.  

  Authors state in the abstract about the expression of POMP, PSMB5, NRF2, XBP1, cMAF and MAF, but in results they only show POMP, NRF2, and MAFb. Abstract needs to be fixed since authors focus are these 3 proteins.  

Response: We have reedited the Abstract according to the reviewer’s suggestion.

Results section needs to be organized into subsections and have its own results, so that readers would follow the manuscript better.

Response: The Results section has been organized into subsections as requested.

Abstract: While multiple myeloma (MM) treatment with proteasome inhibitors and other agents yields encouraging results, primary and secondary resistance remains an emerging problem. An important factor in such treatment resistance is the overexpression of several proteins. The present study comprehensively evaluates the expression of POMP, PSMB5, NRF2, XBP1, cMAF and MAFB proteins in plasma cells isolated from the bone marrow of 39 MM patients treated with bortezomib-based regimens using enzyme-linked immunosorbent assay (ELISA). The proteins were selected on the basis of previous laboratory and clinical studies in bortezomib treated MM patients. Higher levels of cMAF were observed in patients with ISS stage III compared to stage I and II. (p = 0.0383) NRF2 level was increased in patients with complete response or very good partial response response after bortezomib-based therapy compared to those with a worse response (p = 0.0167.  The expression of some proteins correlated with overall survival (OS). Among the proteins that significantly impacted OS in univariate analyses, high expression of POMP (p = 0.0277) was related with shorter OS, whereas high expression of MAFB (p = 0.0147) was associated with longer OS. In conclusion, our results indicate that high expression of POMP and MAFB in MM plasma cells may serve as predictors of OS in MM patients treated with bortezomib-based regimens. However, further studies are needed to determine the role of these factors in effective strategies for improving anti-myeloma therapy.